# Exploring Stable Meta-optimization Patterns via Differentiable Reinforcement Learning for Few-shot Classification

## ABSTRACT

Existing few-shot learning methods generally focus on designing exquisite structures of meta-learners for learning task-specific prior to improve the discriminative ability of global embeddings. However, they often ignore the importance of learning stability in meta-training, making it difficult to obtain a relatively optimal model. From this key observation, we propose an innovative generic differentiable Reinforcement Learning (RL) strategy for few-shot classification. It aims to explore stable meta-optimization patterns in meta-training by learning generalizable optimizations for producing task-adaptive embeddings. Accordingly, our differentiable RL strategy models the embedding procedure of feature transformation layers in meta-learner to optimize the gradient flow implicitly. Also, we propose a memory module to associate historical and current task states and actions for exploring inter-task similarity. Notably, our RL-based strategy can be easily extended to various backbones. In addition, we propose a novel task state encoder to encode task representation, which fully explores inner-task similarities between support set and query set. Extensive experiments verify that our approach can improve the performance of different backbones and achieve promising results against state-of-the-art methods in few-shot classification. Our code is available at an anonymous site: https://anonymous.4open.science/r/db8f0c012/.

## CCS CONCEPTS

• **Computing methodologies → Image representations**.

## KEYWORDS

few-shot classification, differentiable reinforcement learning, stable meta-optimization.

## 1 INTRODUCTION

Few-shot learning [1, 18, 30, 42, 55] aims to explore generalizable visual knowledge from seen classes in training and then utilize the learned knowledge to correctly recognize unseen classes where only a few labeled samples are available. Few-shot learning has attracted ever-increasing interest in industrial and academic communities due to its various applications.

Generally, existing methods [20, 28, 32, 53] often rely on designing exquisite structures of meta-learner for generating discriminative

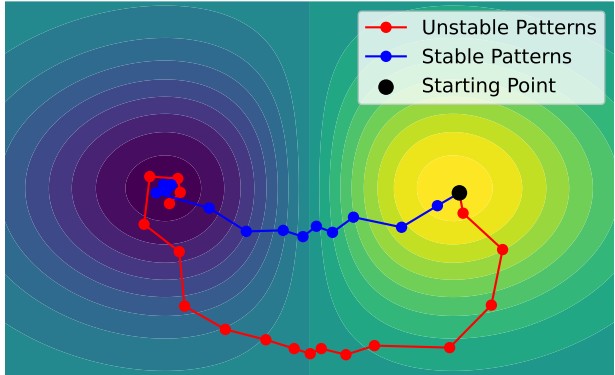

**(a) An example of unstable optimization patterns.**

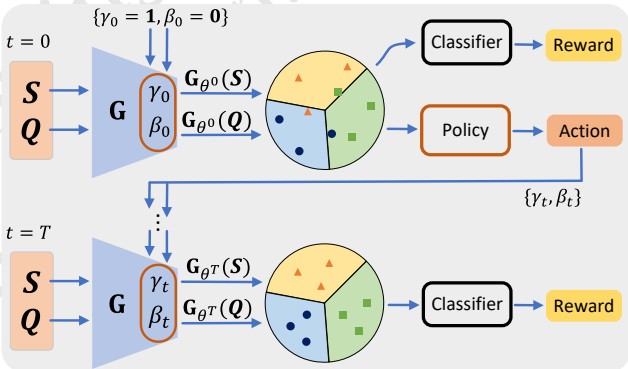

**(b) Overview of our approach.**

**Figure 1: (a) The illustration of our motivation. Unstable patterns of meta-learner can lead to wrong optimization direction, causing extra gradient descent computation. (b) Overview of our differentiable RL strategy for modeling generalizable patterns of producing transformation layer parameters.**

global embeddings of images to achieve promising performance. Recently, exploring task-specific prior for meta-tasks has been proven as an efficient way to generate discriminative global embeddings. To this end, existing methods [5, 36, 55] usually adopt task-adapting strategies to model the task-specific prior. Ye *et al.* [55] modeled the task-adapting strategy as non-linear transformations on the support set. Qiao *et al.* [36] formulated the task-adapting process as a quadratic programming problem on sample-wise distance constraints. Baik *et al.* [5] proposed to implicitly guide the embedding procedure via optimizations of task-specific loss functions. Despite the improvement of these methods, the meta-training process still suffers from severe problems, such as high variance [4] and noisy gradients [43]. These problems can lead to unstable optimization patterns in meta-training and cause wrong optimization direction, finally affecting the performance, as illustrated in Figure 1(a).

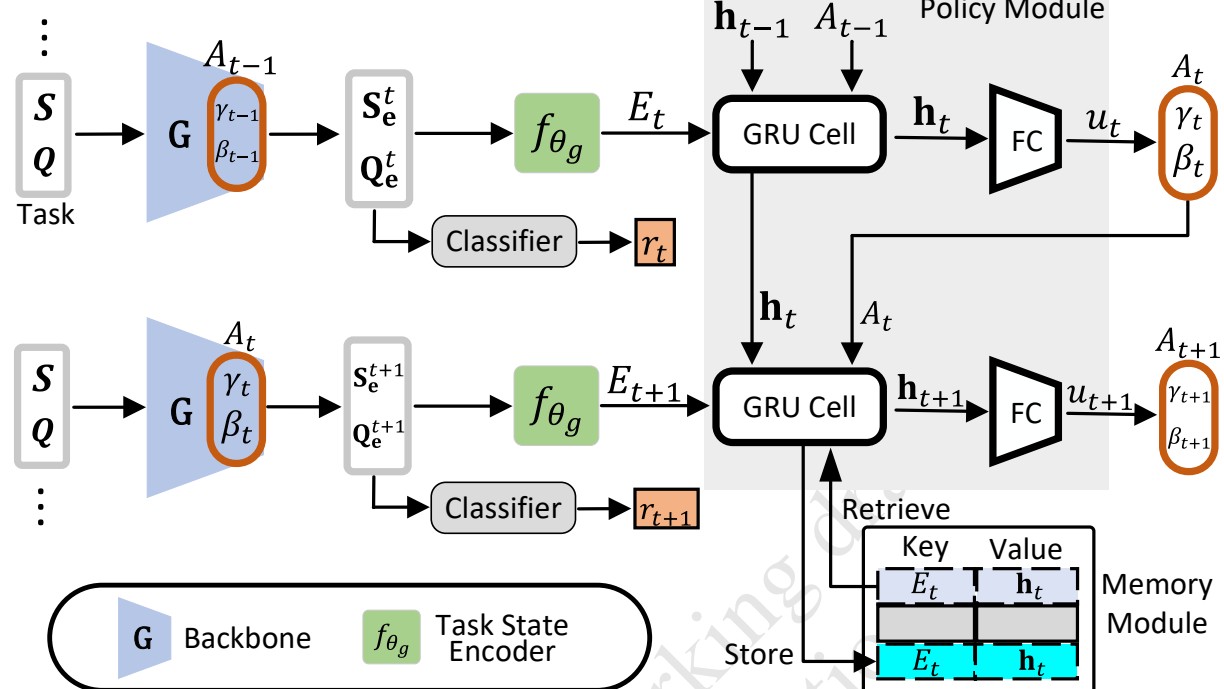

**Figure 2: The details of our approach. We describe the transaction from time step $t$ to $t + 1$. The policy module, consisting of a GRU cell with a FC layer, takes the state of the current time step $t$ as input and outputs the parameters of the transformation layer (*i.e.*, scale $\gamma$ and shift $\beta$) as action $A_t$. The state of the current time step $t$ is the concatenation of the current task state $E_t$ and previous action $A_{t-1}$. The memory module enables rapid storage and retrieval of associations between task state $E_t$ and hidden state $\mathrm{h}_t$ of action $A_t$.**

Prior approaches of reinforcement learning, including training sample augment [9] and spatial attention enhancement [17], have proven their effectiveness in improving the performance of few-shot learning tasks. However, these methods mainly straightforwardly utilize RL in the pre-training stage, and they often suffer from unstable training and huge computational cost. This is because they model the environment as a non-differentiable black box, resulting in the training process requiring lots of trials and errors. Moreover, black-box RL is prone to underfit on unseen tasks outside of distribution [29]. For the meta-training stability, Anantha *et al.* [3] modeled the meta-learner as the agent of RL to learn the task-adaptive loss surface of base-learner for exploiting stable loss patterns. However, Anantha *et al.* [3] aims to stabilize the inner loop. In contrast, their approach is still limited to the outer loop since the environment lacks differentiability. Applying RL with a differentiable environment is proven to be efficient in robotics [29], control [14], and navigation [10], by the ability to stabilize optimization process and improve generalization on unseen tasks. This inspires us to utilize the pros of differentiable RL to overcome the flaws of black-box RL-based approaches (*e.g.*, optimization instability, and huge computational cost) in few-shot learning.

Highly inspired by differentiable RL [14, 29], we propose a generic differentiable RL-based meta-optimization strategy for few-shot classification (named **R**einforced **M**eta-**O**ptimization, **RMO**), as shown in Figure 1(b). Our method aims to explore stable meta-optimization patterns in meta-training for improving the discriminative ability of task-adaptive embeddings. Hence, we design a

differentiable RL policy to model the embedding procedure of transformation layers in meta-learner for implicitly optimizing the gradient flow. Specifically, we model the policy as a Gated Recurrent Unit (GRU) to learn generalizable optimization patterns across meta-tasks. To this end, we propose a memory module, modeled as a Differentiable Neural Dictionary (DND), to associate historical and current task states and actions for exploring inter-task similarity and boosting the learning of policy. In addition, we propose a novel task state encoder to encode task state for fully exploring inner-task similarity between supports and queries. We conduct experiments with various backbones and classifiers on three popular datasets. The results verify that our method can consistently improve the performance on different backbones and achieve promising results against state-of-the-art transductive methods in few-shot classification.

Overall, the contributions are summarized as follows:

(1) We propose a generic differentiable RL-based few-shot learning strategy to explore stable meta-optimization patterns for meta-training. Our approach can be easily extended to various backbones, including ConvNet, ResNet, WRN, and DenseNet.
(2) We propose a novel task state encoder to encode task representation, enabling information adaptation and fully exploring inner-task similarities between support set and query set.
(3) Extensive experiments verify that our approach consistently improves the performance on various backbones and achieves promising results against state-of-the-art transductive methods in few-shot classification.

## 2 RELATED WORK

### 2.1 Few-shot Classification

Meta-learning has been proven to be an effective way to address the few-shot classification problem [18, 24, 30, 57, 59]. These methods often train a meta-learner using diverse meta-tasks sampled from a task distribution to capture common patterns across the tasks. The design of a meta-learner can include learning task-adapted representations [24, 30, 55], exploring inner-task correlations [18, 36], and utilizing text-modal knowledge [32, 51]. Introducing transformation layers into the network has proven efficient in learning task-specific representations. These methods include feature map modulation [30, 46], non-linear adaptation [55], task-level projection [57], and learning task-specific loss functions [5]. While these methods always tend to design complicated structures of meta-learner [17], they ignore the importance of stable optimization patterns in meta-training. These methods often suffer from overfitting in meta-training and need careful parameter tuning [30].

### 2.2 Stable Optimization for Few-shot Learning

MAML [12] is a powerful algorithm introducing meta-learning into diverse problems, including few-shot classification. It aims to learn optimizations to adapt to unseen tasks with few available training samples. However, MAML-based methods suffer from some problems, including training instability [4], permutation sensitivity [54], and memorization overfitting [56]. Solutions for MAML's shortcomings have been well studied. Ye *et al.* [54] proposed to initialize the classification weights with one single weight vector to overcome the permutation sensitivity problem in few-shot classification. Simon *et al.* [43] proposed to scale the noisy gradients of meta-learner via low-rank approximation to stabilize the meta-training. However, these methods mainly focus on improving the generic meta-learning algorithms. In few-shot learning, the stable optimization patterns in meta-training need further studying.

### 2.3 Reinforced Few-shot Learning

Reinforcement Learning has been proven effective in addressing the few-shot learning problem. Chu *et al.* [9] proposed to utilize maximum-entropy RL to learn the cropping trajectories of training images. Hong *et al.* [17] proposed to utilize the RL strategy to optimize spatial attention of feature maps. However, these methods only utilized the RL strategy into the pre-training stage, and these methods are still limited by the shortcomings of RL (*e.g.* unstable training and huge computational cost). On training stability of few-shot learning, Anantha *et al.* [3] proposed to learn the scale of gradients of base-learner task-dependently via RL to explore stable loss patterns in meta-training. However, this work only studied the optimization stability of the inner loop in meta-training, and for the outer loop, this method is still limited by the shortcomings of RL since the environment is modeled as a black box.

## 3 METHOD

Our approach (as shown in Figure 2) mainly consists of a policy module, a memory module, and a task state encoder. We describe each module and the optimization strategy of our whole method in the following subsections.

**Problem statement.** In the meta-training phase, the model faces a set of tasks. Each task is represented as a $N$-way $K$-shot problem with $N$ classes sampled from the seen class set $\mathbb{S}$ (*i.e.*, classes in training set) and $K$ labeled samples per class, which composes the support set. Besides, $M$ unlabeled samples are sampled per class as the query set. We denote the support set as $\mathbf{S} = \{\mathbf{x}_i, \mathbf{y}_i\}_{i=1}^{NK}$ and the query set as $\mathbf{Q} = \{\mathbf{x}_i\}_{i=NK+1}^{NK+NM}$ with the instance $\mathbf{x}_i$ and the label $\mathbf{y}_i \in \mathbb{S}$. In validation and evaluation, the $N$ classes are sampled from the unseen class set $\mathbb{U}$ (*i.e.*, classes in the validation set and testing set). Note that $\mathbb{S} \cap \mathbb{U} = \emptyset$. The goal is to find an optimal function $f$ that correctly classifies the test sample $\mathbf{x}_{test}$ via $\hat{\mathbf{y}}_{test} = f(\mathbf{x}_{test}; \mathbf{S}) \in \mathbb{U}$.

**Model design.** First, we describe how the Feature-wise Linear Modulation (FiLM) layer [33] transforms the feature map. The learnable parameters of a FiLM layer include scale $\gamma$ and shift $\beta$, where $\gamma, \beta \in \mathbb{R}^c$. We split the backbone network $\mathbf{G}$ into two parts, where $\mathbf{G}_1$ is the unchanged part and $\mathbf{G}_2$ is the part where a FiLM layer is inserted after the last Batch Normalization (BN) layer in each convolution block. An intermediate feature map $\mathbf{e}'_\mathbf{x} = \mathbf{G}_1(\mathbf{x})$ of an input image $\mathbf{x}$ is first obtained. Denote $\mathbf{z} \in \mathbb{R}^{h*w*c}$ is the activation of $\mathbf{e}'_\mathbf{x}$ produced by the BN layer in $\mathbf{G}_2$. Then $\mathbf{z}$ is modulated as: $\hat{z}_{h,w,c} = \gamma_c * z_{h,w,c} + \beta_c$, where $\hat{z}_{h,w,c} \in \hat{\mathbf{z}}$, $z_{h,w,c} \in \mathbf{z}$, and $c$ denotes the number of channels. This modulation is applied in each block of $\mathbf{G}_2$. At last, the part $\mathbf{G}_2$ outputs the embedding $\mathbf{e}_\mathbf{x} \in \mathbb{R}^d$ of input image $\mathbf{x}$, denoted as $\mathbf{e}_\mathbf{x} = \mathbf{G}_2(\mathbf{e}'_\mathbf{x}; \gamma, \beta)$. Denote $\theta_\mathbf{G}$ as the set of all learnable parameters of $\mathbf{G}$. Different from previous methods [30, 39], we model the learning process of parameters $\gamma$ and $\beta$ with differentiable RL to optimize the gradient flow of the meta-learner. We describe how we model this process with the key elements of differentiable RL in the following.

**State.** RL is an iterative process where the policy takes the current state as input and picks up an action in each time step. Denote $t = 1, \cdots, T$ as the current time step and $T$ as the total number of time steps. The next state $S_{t+1}$ is obtained from a differentiable transition function: $S_{t+1} = \Phi(S_t, A_t)$. We model the state $S_t$ as the concatenation of current task state $E_t$ and previous action $A_{t-1}$. We depict how we obtain the task state $E_t$ via task state encoder in **Section 3.3. Task State Encoder**.

**Action.** In each time step, the policy outputs an action based on the current task state. We model the action $A_t$ as the concatenation of parameters $\gamma_t$ and $\beta_t$. Here $\gamma_t$ and $\beta_t$ denotes the values of $\gamma$ and $\beta$ in time step $t$, respectively. We depict how the action $A_t$ is picked up in **Section 3.1. Policy Module**.

**Reward.** The environment rewards the quality of the action to guide the RL model. The reward $r_t$ is calculated with a differentiable function: $r_t = \Psi(S_t, A_t)$. We model the current reward $r_t$ as the classification loss calculated by the classifier. We depict how the reward $r_t$ is calculated in **Section 3.4. Optimization Strategy**.

### 3.1 Policy Module

We model the policy module as a minimal gated unit [60], a variant of GRU with one single forget gate, to improve efficiency and reduce the number of parameters. In each time step $t$, the embeddings $\{\mathbf{e}_\mathbf{x}^t, \mathbf{x} \in \mathbf{S} \cup \mathbf{Q}\}$ are first transformed into the task state $E_t$, then concatenated with previous action $A_{t-1}$ to form input state $S_t$. Define $\gamma_0 = \mathbf{1}$ and $\beta_0 = \mathbf{0}$, where $\mathbf{1}$ and $\mathbf{0}$ are two vectors filled with 1 and 0. The GRU cell takes $S_t$ as input and outputs the action $A_t = \{\gamma_t, \beta_t\}$

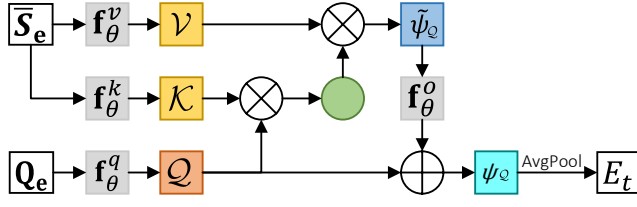

**Figure 3: Overview of the task state encoder, which consists of a self-attention module followed by an average pooling layer.**

via the following equations:

$$
\begin{aligned}
\mathbf{f}_t &= \sigma(\mathbf{W}_f \cdot [\mathbf{h}_{t-1}, S_t] + \mathbf{b}_f), \\
\tilde{\mathbf{h}}_t &= \tanh(\mathbf{W}_h \cdot [\mathbf{f}_t \odot \mathbf{h}_{t-1}, S_t] + \mathbf{b}_h), \\
\mathbf{h}_t &= (1 - \mathbf{f}_t) \odot \mathbf{h}_{t-1} + \mathbf{f}_t \odot \tilde{\mathbf{h}}_t,
\end{aligned}
\tag{1}
$$

where $\sigma(\cdot)$ and $\tanh(\cdot)$ denote the sigmoid and hyperbolic tangent activation function, respectively, and $\odot$ denotes the Hadamard product. Then $\mathbf{h}_t$ is taken by a fully connected layer without bias to get the mean $\mu_t$: $\mu_t = \mathbf{W}_u \cdot \mathbf{h}_t$. The action $A_t$ is sampled from the normal distribution: $A_t \sim \mathcal{N}(\mu_t, 1)$. The value of $\mathbf{h}_0$ is initialized to $\mathbf{0}$ for each task.

### 3.2 Memory Module

Inspired by the idea of episodic control [34, 40], we model the memory module as a DND to explore inter-task similarity, enabling rapid storage and retrieve of associations between task state $E_t$ and hidden state $\mathbf{h}_t$ of action $A_t$. The key and value of DND denote task state $E_t$ and hidden state $\mathbf{h}_t$, respectively. Then the new $\mathbf{h}_t$ is calculated using the following equations:

$$
\begin{aligned}
\mathbf{f}_t &= \sigma(\mathbf{W}_f \cdot [\mathbf{h}_{t-1}, S_t] + \mathbf{b}_f), \\
\tilde{\mathbf{h}}_t &= \tanh(\mathbf{W}_h \cdot [\mathbf{f}_t \odot \mathbf{h}_{t-1}, S_t] + \mathbf{b}_h), \\
\mathbf{r}_t &= \tanh(\mathbf{W}_r \cdot [\mathbf{h}_{t-1}, S_t] + \mathbf{b}_r), \\
\mathbf{h}_t &= (1 - \mathbf{f}_t) \odot \mathbf{h}_{t-1} + \mathbf{f}_t \odot \tilde{\mathbf{h}}_t + \mathbf{r}_t \odot \tanh(\mathbf{h}_{ep}),
\end{aligned}
\tag{2}
$$

where $\mathbf{h}_{ep}$ denotes the hidden state from DND: $\mathbf{h}_{ep} = DND\left[E'_t\right]$, and $E'_t$ denotes the matched key of current $E_t$ using a distance metric. We choose $E'_t$ using top-1 cosine similarity with stored $E_t$: $d(E'_t, E_t) = \frac{E'_t \cdot E_t}{\|E'_t\| \|E_t\|}$. Finally, $\mathbf{h}_t$ and $E_t$ are stored to DND: $DND[E_t] = \mathbf{h}_t$. Denote $\theta_p$ as the set of all learnable parameters of policy module and $D$ as the number of memory units in DND.

### 3.3 Task State Encoder

The architecture of the task state encoder is shown in Figure 3. Unlike previous methods [30, 39] that only consider inner-task information from the support set, we utilize a single-head self-attention module to explore inner-task similarities between support set and query set. Here, we omit time step $t$ for simplicity. First, we obtain class representations by averaging the support samples in each class: $\bar{\mathbf{e}}_c = \frac{1}{K} \sum_{(\mathbf{x}_i, \mathbf{y}_i) \in \mathbf{S}_c} \mathbf{e}_{\mathbf{x}_i}$, where $\mathbf{S}_c$ denotes the support samples in $c$-th class, and $c = 1, \cdots, N$. Define $\bar{\mathbf{S}}_\mathbf{e} = \{\bar{\mathbf{e}}_c\}_{c=1}^{N}$ and $\mathbf{Q}_\mathbf{e} = \{\mathbf{e}_{\mathbf{x}_i}\}_{\mathbf{x}_i \in \mathbf{Q}}$. We construct the query, key, and value using the following equation:

$$
Q = \mathbf{f}_\theta^q(\mathbf{Q_e}), \ \mathcal{K} = \mathbf{f}_\theta^k(\bar{\mathbf{S}}_\mathbf{e}), \ \mathcal{V} = \mathbf{f}_\theta^v(\bar{\mathbf{S}}_\mathbf{e}),
\tag{3}
$$

**Algorithm 1** Training procedure of our approach.

---
**Require:** Seen class set $\mathbb{S}$, classifier $f_m$
1: **while** *training* **do**
2:     Sample $N$-way $K$-shot task $(\mathbf{S}, \mathbf{Q})$ from $\mathbb{S}$
3:     Compute $\mathbf{e}_\mathbf{x}^0 = G(\mathbf{x}; \gamma_0, \beta_0)$ for all $\mathbf{x} \in \mathbf{S} \cup \mathbf{Q}$
4:     Compute $L_c$ for all $\mathbf{x}_{test} \in \mathbf{Q}$ with $f_m$
5:     Reset $\mathbf{h}_0 = \mathbf{0}$
6:     **for all** $t = 1, \cdots, T$ **do**
7:         Compute $S_t = \{E_t, \gamma_{t-1}, \beta_{t-1}\}$ for all $\mathbf{e}_\mathbf{x}^{t-1}$
8:         Compute $A_t = \{\gamma_t, \beta_t\}$ for $S_t$
9:         Compute $\mathbf{e}_\mathbf{x}^t = G(\mathbf{x}; \gamma_t, \beta_t)$ for all $\mathbf{x} \in \mathbf{S} \cup \mathbf{Q}$
10:       Compute $r_t$ for all $\mathbf{x}_{test} \in \mathbf{Q}$ with $f_m$
11:     **end for**
12:     Compute $L_r$ for all $\{r_t\}_{t=1}^T$
13:     Update parameters $\theta_\mathbf{G}, \theta_p,$ and $\theta_g$ with $L_{total}$
14: **end while**
15: **return** Parameters $\theta_\mathbf{G}, \theta_p,$ and $\theta_g$
---

where the support classes and query samples are mapped by three linear projections, $\mathbf{f}_\theta^q$, $\mathbf{f}_\theta^k$, and $\mathbf{f}_\theta^v$, respectively. The calculation of task state $E$ is formulated as:

$$
\begin{aligned}
\tilde{\psi_Q} &= softmax(\frac{Q\mathcal{K}^T}{\sqrt{d}})\mathcal{V}, \\
\psi_Q &= \tau(Q + \mathbf{f}_\theta^o(\tilde{\psi_Q})), \\
E &= AvgPool(\psi_Q),
\end{aligned}
\tag{4}
$$

where $\mathbf{f}_\theta^o$ is a linear projection and $\tau$ denotes the sequence of dropout [45] and layer normalization [52]. Finally, $E$ is calculated via the average pooling operation, denoted by $AvgPool(\cdot)$. Denote $\theta_g$ as the set of all learnable parameters of the task state encoder.

### 3.4 Training Strategy

We choose 3 popular classifiers: ProtoNet [44], DSN [42], and MetaOptNet [23]. The corresponding models are named RMO-PN, RMO-DSN, and RMO-MON, respectively. Denote the predicted probability distribution of test sample $(\mathbf{x}_q, \mathbf{y}_q)$ as $p(\hat{\mathbf{y}}_q = \mathbf{y}_q \mid \mathbf{e}_{\mathbf{x}_q}; f_m)$, where $f_m$ denotes the classifier. We first define the classification loss as:

$$
L_c = -\log p(\hat{\mathbf{y}}_q = \mathbf{y}_q \mid G(\mathbf{x}_q; \gamma_0, \beta_0)).
\tag{5}
$$

Then, in each time step $t$, the log-probability is calculated as the reward:

$$
r_t = \log p(\hat{\mathbf{y}}_q = \mathbf{y}_q \mid G(\mathbf{x}_q; \gamma_t, \beta_t)).
\tag{6}
$$

The goal is to maximize the accumulated rewards $\{r_t\}_{t=1}^T$. Denote $\gamma$ as the discount factor. We define the reinforcement loss as:

$$
L_r = \sum_{t=1}^{T} \log \pi(A_t \mid S_t) R_t,
\tag{7}
$$

where $R_t = \sum_{k=0}^{T-t} \gamma^k r_{t+k}$ denotes a discounted accumulated reward, $\pi \sim \mathcal{N}(\mu_t, 1)$. The total loss is defined as:

$$
L_{total} = L_c + \rho L_r,
\tag{8}
$$

where $\rho$ is a scalar hyper-parameter. The pseudocode of our approach is described in Algorithm 1.

**Table 1: Few-shot *5-way* classification accuracy and 95% confidence interval on *Mini*ImageNet dataset.**

| Model | Backbone | 1-shot | 5-shot |
|---|---|---|---|
| ProtoNet[44] | Conv-4 | 52.61±0.20 | 71.33±0.16 |
| TEAM[36] | Conv-4 | 56.57 | 72.04 |
| EPNet[41] | Conv-4 | 59.32±0.88 | 72.95±0.64 |
| Curvature+T[13] | Conv-4 | 58.29±0.22 | 73.93±0.16 |
| DSN-MR[42] | Conv-4 | 55.88±0.90 | 70.50±0.68 |
| ALFA+MeTAL[5] | Conv-4 | 57.75±0.38 | 74.10±0.43 |
| AIM[22] | Conv-4 | 61.90±0.57 | 74.55±0.38 |
| **RMO-PN** | Conv-4 | 61.50±0.20 | 75.11±0.16 |
| **RMO-MON** | Conv-4 | 60.15±0.20 | 74.64±0.18 |
| **RMO-DSN** | Conv-4 | **62.52±0.20** | **76.32±0.16** |
| ProtoNet[44] | ResNet-12 | 62.39±0.21 | 80.53±0.14 |
| MetaOptNet[23] | ResNet-12 | 62.64±0.61 | 78.63±0.46 |
| DSN-MR[42] | ResNet-12 | 64.60±0.72 | 79.51±0.50 |
| CAN+T[18] | ResNet-12 | 67.19±0.55 | 80.64±0.35 |
| Curvature+T[13] | ResNet-12 | 71.79±0.23 | 83.00±0.17 |
| ALFA+MeTAL[5] | ResNet-12 | 66.61±0.28 | 84.40±0.44 |
| LR+ICI[50] | ResNet-12 | 66.80 | 79.26 |
| EPNet[41] | ResNet-12 | 66.50±0.89 | 81.06±0.60 |
| **RMO-PN** | ResNet-12 | 71.21±0.20 | 86.03±0.15 |
| **RMO-MON** | ResNet-12 | 69.71±0.22 | 85.52±0.15 |
| **RMO-DSN** | ResNet-12 | **73.74±0.20** | **86.69±0.15** |
| TEAM[36] | ResNet-18 | 60.07 | 75.90 |
| TIM-GD[7] | ResNet-18 | 73.9 | 85.0 |
| LaplacianShot[61] | ResNet-18 | 72.11±0.19 | 82.31±0.14 |
| Oblique+T[35] | ResNet-18 | 77.20±0.36 | 87.11±0.42 |
| **RMO-PN** | ResNet-18 | 75.31±0.20 | 86.32±0.15 |
| **RMO-DSN** | ResNet-18 | **77.85±0.20** | **87.54±0.16** |
| BD-CSPN[26] | WRN-28-10 | 70.31±0.93 | 81.89±0.60 |
| AIM[22] | WRN-28-10 | 71.22±0.57 | 82.25±0.34 |
| LaplacianShot[61] | WRN-28-10 | 74.86±0.19 | 84.13±0.14 |
| EPNet[41] | WRN-28-10 | 70.74±0.85 | 84.34±0.53 |
| TIM-GD[7] | WRN-28-10 | 77.8 | 87.4 |
| Oblique+T[35] | WRN-28-10 | 80.64±0.34 | 89.39±0.39 |
| **RMO-PN** | WRN-28-10 | 79.27±0.19 | 87.81±0.16 |
| **RMO-DSN** | WRN-28-10 | **81.08±0.20** | **89.97±0.16** |

**Table 2: Few-shot *5-way* classification accuracy and 95% confidence interval on *Tiered*ImageNet dataset.**

| Model | Backbone | 1-Shot | 5-Shot |
|---|---|---|---|
| ProtoNet[44] | Conv4 | 52.28±0.21 | 71.34±0.18 |
| TPN[27] | Conv-4 | 59.91±0.94 | 73.30±0.75 |
| EPNet[41] | Conv-4 | 59.97±0.95 | 73.91±0.75 |
| ALFA+MeTAL[5] | Conv-4 | 60.29±0.37 | 75.88±0.29 |
| **RMO-PN** | Conv-4 | 62.10±0.20 | 77.80±0.19 |
| **RMO-MON** | Conv-4 | 61.82±0.21 | 76.24±0.19 |
| **RMO-DSN** | Conv-4 | **63.39±0.20** | **78.84±0.18** |
| ProtoNet[44] | ResNet-12 | 68.23±0.23 | 84.03±0.16 |
| MetaOptNet[23] | ResNet-12 | 65.99±0.72 | 81.56±0.53 |
| DSN-MR[42] | ResNet-12 | 67.39±0.82 | 82.85±0.56 |
| CAN+T[18] | ResNet-12 | 73.21±0.58 | 84.93±0.38 |
| Curvature+T[13] | ResNet-12 | 77.19±0.24 | 86.18±0.15 |
| ALFA+MeTAL[5] | ResNet-12 | 70.29±0.40 | 86.17±0.35 |
| EPNet[41] | ResNet-12 | 76.53±0.87 | 87.32±0.64 |
| LR+ICI[50] | ResNet-12 | 80.79 | 87.92 |
| **RMO-PN** | ResNet-12 | 80.29±0.23 | 87.54±0.16 |
| **RMO-MON** | ResNet-12 | 79.44±0.25 | 87.09±0.19 |
| **RMO-DSN** | ResNet-12 | **81.11±0.23** | **88.22±0.16** |
| TIM-GD[7] | ResNet-18 | 79.9 | 88.5 |
| LaplacianShot[61] | ResNet-18 | 78.98±0.21 | 86.39±0.16 |
| Oblique+T[35] | ResNet-18 | 83.73±0.36 | 90.46±0.46 |
| **RMO-PN** | ResNet-18 | 82.05±0.23 | 88.67±0.16 |
| **RMO-DSN** | ResNet-18 | **84.03±0.23** | **90.86±0.16** |
| BD-CSPN[26] | WRN-28-10 | 78.74±0.95 | 86.92±0.63 |
| LaplacianShot[61] | WRN-28-10 | 80.18±0.21 | 87.56±0.15 |
| EPNet[41] | WRN-28-10 | 78.50±0.91 | 88.36±0.57 |
| TIM-GD[7] | WRN-28-10 | 82.1 | 89.8 |
| Oblique+T[35] | WRN-28-10 | 85.22±0.34 | 91.35±0.42 |
| **RMO-PN** | WRN-28-10 | 83.66±0.22 | 89.23±0.18 |
| **RMO-DSN** | WRN-28-10 | **85.54±0.22** | **91.72±0.18** |

## 4 EXPERIMENTS

### 4.1 Datasets

We evaluate our approach on 3 benchmark datasets: *Mini*ImageNet [47], *Tiered*ImageNet [38], and CUB-200-2011 [48]. *Mini*ImageNet and *Tiered*ImageNet are two subsets of ImageNet [11] dataset. *Mini*ImageNet dataset contains 100 categories with 600 images per category. Following the splits provided by Ravi *et al.* [37], we split the 100 categories into 64, 16, and 20 categories for training, validation, and evaluation, respectively. *Tiered*ImageNet is a large-scale dataset containing 351, 97, and 160 categories for training, validation, and evaluation, respectively. The CUB-200-2011 dataset contains 200 categories with a total number of 11788 images. Following the protocol of Hilliard *et al.* [16], we split the dataset into 100, 50, and 50 categories for training, validation, and evaluation, respectively.

### 4.2 Implementation Details

We utilize 4 commonly used backbones in few-shot learning: Conv-4 [21], ResNet-12 [15], ResNet-18 [15], and WRN-28-10 [58]. The backbone takes a 3×84×84 image as input. For Conv-4, the Adam optimizer is utilized. For ResNets and WRN, the SGD optimizer is utilized with momentum of 0.9 and weight decay of 0.0005. We insert a FiLM layer after the last BN layer in the convolution blocks of the backbone. The parameters $\gamma_t$ and $\beta_t$ of each FiLM layer are concatenated to form the action $A_t$. The value of $T$, $D$, and $\gamma$ is set

**Table 3: Few-shot *5-way* classification accuracy and 95% confidence interval on the CUB-200-2011 dataset.**

| Model | Backbone | 1-Shot | 5-Shot |
|---|---|---|---|
| ProtoNet[44] | Conv-4 | 63.72±0.22 | 81.50±0.15 |
| EPNet[41] | Conv-4 | 65.94±0.93 | 78.80±0.64 |
| TEAM[36] | Conv-4 | 75.71 | 86.04 |
| BD-CSPN[26] | Conv-4 | 75.10 | 87.25 |
| Curvature+T[13] | Conv-4 | 76.69±0.21 | **89.30±0.12** |
| **RMO-PN** | Conv-4 | 75.96±0.23 | 87.56±0.15 |
| **RMO-MON** | Conv-4 | 74.28±0.25 | 85.54±0.17 |
| **RMO-DSN** | Conv-4 | **77.26±0.24** | 88.73±0.16 |
| ProtoNet[44] | ResNet-12 | 66.09±0.92 | 82.50±0.58 |
| BD-CSPN[26] | ResNet-12 | 84.90 | 90.22 |
| EPNet[41] | ResNet-12 | 82.85±0.81 | 91.32±0.41 |
| LR+ICI[50] | ResNet-12 | **88.06** | 92.53 |
| **RMO-PN** | ResNet-12 | 84.60±0.21 | 91.39±0.12 |
| **RMO-MON** | ResNet-12 | 83.32±0.26 | 90.31±0.18 |
| **RMO-DSN** | ResNet-12 | 86.79±0.21 | **92.82±0.13** |
| ProtoNet[44] | ResNet-18 | 72.99±0.88 | 86.65±0.51 |
| TEAM[36] | ResNet-18 | 80.16 | 87.17 |
| TIM-GD[7] | ResNet-18 | 82.2 | 90.8 |
| LaplacianShot[61] | ResNet-18 | 80.96 | 88.68 |
| Oblique+T[35] | ResNet-18 | 85.87 | 94.97 |
| **RMO-PN** | ResNet-18 | 86.46±0.23 | 93.69±0.14 |
| **RMO-DSN** | ResNet-18 | **87.57±0.23** | **95.16±0.14** |
| BD-CSPN[26] | WRN-28-10 | 87.45 | 91.74 |
| EPNet[41] | WRN-28-10 | 87.75±0.70 | 94.03±0.33 |
| **RMO-PN** | WRN-28-10 | 86.76±0.22 | 93.67±0.13 |
| **RMO-DSN** | WRN-28-10 | **88.56±0.22** | **95.27±0.13** |

**Table 4: Few-shot *5-way* classification accuracy and 95% confidence interval with DenseNet backbone.**

| Method | *Mini*ImageNet | | *Tiered*ImageNet | |
|---|---|---|---|---|
| | 1-shot | 5-shot | 1-shot | 5-shot |
| SimpleShot[49] | 65.77±0.19 | 82.23±0.13 | 71.20±0.22 | 86.33±0.15 |
| LaplacianShot[61] | 75.57±0.19 | 84.72±0.13 | 80.30±0.20 | 87.93±0.15 |
| RAP-LaplacianShot[17] | 75.58±0.20 | 85.63±0.13 | - | - |
| ICA+MSP[25] | 77.06±0.26 | 84.99±0.14 | 84.29±0.25 | 89.31±0.15 |
| **RMO-PN** | 78.26±0.19 | 84.92±0.14 | 85.15±0.21 | 89.25±0.15 |
| **RMO-DSN** | **79.83±0.19** | **86.47±0.13** | **86.09±0.20** | **91.16±0.15** |

to 5, 2000, and 0.9, respectively. The value of $\rho$ is set to $\frac{2}{\sum_{i=1}^{T} i}$. First, we pre-train the backbone following the settings of the prior work [55]. Then, the whole model is meta-trained for 200 epochs, with each epoch including 100 randomly sampled tasks. We scale the learning rate of weights of the backbone by 0.1 in meta-training. In validation, we use 600 randomly sampled tasks to choose the best model. During evaluation, we report the classification accuracy and 95% interval confidence over 10000 randomly sampled tasks. Each

**Table 5: Few-shot *5-way* classification accuracy comparing to CNAPS and Simple-CNAPS. "In." denotes "inductive".**

| Method | *Mini*ImageNet | | *Tiered*ImageNet | |
|---|---|---|---|---|
| | 1-shot | 5-shot | 1-shot | 5-shot |
| CNAPS[39] | 77.99 | 87.31 | 75.12 | 86.57 |
| Simple-CNAPS[6] | 82.16 | 89.80 | 78.29 | 89.01 |
| **RMO-PN (In.)** | 80.93 | 89.21 | 77.75 | 89.25 |
| **RMO-DSN (In.)** | **83.12** | **90.76** | **79.35** | **90.16** |

task includes 15 query samples per class in all settings. We take the averaged results using 3 different random seeds. The learning rate is set to 0.002 and 0.001 for ConvNet and three ResNets, respectively. We scale the initial learning rate by 0.2 after every 30 epochs in meta-training. Our code is based on PyTorch [31], and all experiments are conducted using an NVIDIA RTX A6000 GPU. More details are provided in **Supplementary Material**.

## 4.3 Main Results

Table 1, Table 2, and Table 3 list the results of our method on *Mini*ImageNet, *Tiered*ImageNet, and CUB-200-2011, respectively. We mainly compare our approach with the previous state-of-the-art transductive few-shot learning approaches, including Curvature [13], TEAM [36], Oblique [35], EPNet [41], TIM [7], ICI [50], LaplacianShot [61], and BD-CSPN [26]. It can be seen that our approach shows consistent improvements over other transductive few-shot learning approaches on all three datasets using the four backbones. For example, under 1-shot and 5-shot on *Mini*ImageNet with ResNet-12, our approach shows on average +2.3%, +5.8%, and +4.2% performance gains comparing to Curvature [13], CAN [18], and MeTAL [5], respectively. On *Tiered*ImageNet with ResNet-12, our approach shows on average +3.0%, +5.6%, and +6.4% performance gains comparing to the above three methods. On CUB-200-2011, our approach also shows on average +2.9% and +4.8% performance gains with TIM [7] and EPNet [41], respectively. The promising results suggest that our approach makes it easier to reach a better local optima on the loss surface of the meta-training phase via exploring stable meta-optimization patterns, thus improving the discriminative ability of global embeddings.

## 4.4 Ablation Study

In this section, we first explore hyper-parameters' impacts on the proposed strategy. Then, we analyze the impacts of different modules on classification performance. Besides, we extend our strategy to the DenseNet backbone. Furthermore, we conduct experiments on the challenging cross-domain and generalized few-shot tasks. For simplicity, "PN", "DSN", and "MON" denote RMO-PN, RMO-DSN, and RMO-MON, respectively.

**Selection of hyper-parameters.** We describe how we choose the value of hyper-parameter $T$, $D$, and $\gamma$ in **Supplementary Material**.

**Impacts of different modules.** We analyze the impact of each module, *i.e.*, policy, memory, and task state encoder (denoted as "GRU", "DND", and "SATT", respectively). The experiments are conducted under 5-shot 5-way on *Mini*ImageNet. Relative results are listed in Table 6. To show the effects of differentiable RL and memory mechanism, we replace GRU and DND with two FC layers

**Table 6: Classification accuracy and 95% confidence interval on different combinations of the modules in our approach.**

| | SATT | GRU | DND | Conv-4 | | | ResNet-12 | | | ResNet-18 | |
|---|---|---|---|---|---|---|---|---|---|---|---|
| | | | | PN | DSN | MON | PN | DSN | MON | PN | DSN |
| Settings | ✓ | | | 72.33±0.16 | 73.29±0.18 | 71.27±0.17 | 82.76±0.14 | 83.16±0.18 | 82.19±0.18 | 84.74±0.15 | 84.66±0.15 |
| | | ✓ | | 73.18±0.16 | 73.60±0.17 | 72.21±0.17 | 83.17±0.15 | 83.72±0.19 | 83.30±0.17 | 84.39±0.15 | 85.30±0.15 |
| | | ✓ | ✓ | 73.86±0.16 | 74.42±0.17 | 73.03±0.17 | 83.91±0.15 | 84.63±0.20 | 84.21±0.17 | 84.91±0.15 | 85.86±0.15 |
| | ✓ | ✓ | | 74.61±0.16 | 75.64±0.17 | 73.81±0.17 | 85.37±0.15 | 85.86±0.19 | 84.97±0.17 | 85.65±0.15 | 86.73±0.15 |
| | ✓ | ✓ | ✓ | 75.11±0.16 | 76.32±0.16 | 74.64±0.18 | 86.03±0.15 | 86.69±0.15 | 85.52±0.15 | 86.32±0.15 | 87.54±0.16 |

**Table 7: Few-shot *5-way* classification accuracy and 95% confidence interval on cross-domain task: *Mini*ImageNet → CUB-200-2011 dataset.**

| Setups | Backbone | **1-Shot** | **5-Shot** |
|---|---|---|---|
| ProtoNet[44] | ResNet-18 | - | 62.02±0.70 |
| SimpleShot[49] | ResNet-18 | 48.56 | 65.63 |
| MetaOptNet[23] | ResNet-12 | 44.79±0.75 | 64.98±0.68 |
| LaplacianShot[61] | ResNet-18 | 55.46 | 66.33 |
| ALFA+MeTAL[5] | ResNet-12 | - | 70.22±0.14 |
| Centroid[2] | ResNet-18 | 46.85±0.75 | 70.37±1.02 |
| TIM-GD[7] | ResNet-18 | - | 71.0 |
| Oblique+T[35] | ResNet-18 | - | 74.11 |
| **RMO-PN** | ResNet-18 | 54.67±0.18 | 73.58±0.18 |
| **RMO-DSN** | ResNet-18 | **56.82±0.18** | **74.64±0.16** |

**Table 8: Classification accuracy and 95% confidence interval on generalized few-shot task. "In." denotes "inductive".**

| Setup | Model | **SEEN** | **UNSEEN** | **COMBINED** |
|---|---|---|---|---|
| **1-shot** | ProtoNet[44] | 41.73±0.03 | 48.64±0.20 | 35.69 ±0.03 |
| | FEAT[55] | 43.94±0.03 | 49.72±0.20 | 40.50 ±0.03 |
| | **RMO-PN (In.)** | 48.00±0.04 | 53.68±0.19 | 43.61 ±0.04 |
| | **RMO-MON (In.)** | 46.96±0.04 | 52.21±0.19 | 42.80 ±0.03 |
| | **RMO-DSN (In.)** | **49.34±0.04** | **54.16±0.19** | **44.02 ±0.03** |
| **5-shot** | ProtoNet[44] | 41.06±0.03 | 64.94±0.17 | 38.04 ±0.02 |
| | FEAT[55] | 44.94±0.03 | 65.33±0.16 | 41.68 ±0.03 |
| | **RMO-PN (In.)** | 49.56±0.03 | 68.37±0.19 | 44.52 ±0.05 |
| | **RMO-MON (In.)** | 48.71±0.03 | 67.54±0.18 | 43.50 ±0.04 |
| | **RMO-DSN (In.)** | **50.40±0.03** | **69.02±0.17** | **45.13 ±0.04** |

to learn the parameters $\gamma$ and $\beta$. To show the effects of information adaptation from query set when constructing task state, we replace SATT with an averaging operation, *i.e.*, obtaining the task state via averaging the mean vectors represented by each class as Oreshkin *et al.* [30]. It can be observed that the differentiable RL strategy and memory mechanism bring about +3% performance improvement, comparing to learning FiLM layer parameters with FC layers. Besides, the information adaptation from query set also shows about +2% performance improvement.

**Table 9: Time complexity of the three models.**

| Models | Classifier | Total |
|---|---|---|
| RMO-PN | $O(NKd)$ | $O(T(NK)^2d)$ |
| RMO-DSN | $O(NKd^2)$ | $O(max(TNKd^2, T(NK)^2d))$ |
| RMO-MON | $O(d^3)$ | $O(Td^3)$ |

**Table 10: Standard deviations of the loss function.**

| Setting → | 5-way 1-shot | | | | | |
|---|---|---|---|---|---|---|
| Backbone | PN | DSN | MON | RMO-PN | RMO-DSN | RMO-MON |
| Conv-4 | 0.266 | 0.308 | 0.251 | **0.128** | 0.165 | 0.191 |
| ResNet-12 | 0.108 | 0.101 | 0.178 | 0.036 | **0.016** | 0.053 |
| ResNet-18 | 0.127 | 0.102 | 0.201 | 0.042 | **0.015** | 0.041 |
| Setting → | 5-way 5-shot | | | | | |
| Backbone | PN | DSN | MON | RMO-PN | RMO-DSN | RMO-MON |
| Conv-4 | 0.382 | 0.405 | 0.252 | **0.158** | 0.168 | 0.183 |
| ResNet-12 | 0.059 | 0.085 | 0.108 | 0.037 | **0.016** | 0.052 |
| ResNet-18 | 0.054 | 0.035 | 0.067 | 0.034 | **0.013** | 0.041 |

**Time complexity.** The time complexity in each time step for SATT, GRU, and DND are $O((NK)^2d)$, $O(d^2)$, and $O(Dd)$, respectively. The total time complexities of the three models are listed in Table 9. Here we assume $d^3 \gg (NK)^2 \gg D \gg d$.

**Standard deviation of loss function.** The standard deviation of the loss function can verify the stabilizing effects of our approach. We calculate the standard deviation of the loss function in the last 100 epochs. The experiments are conducted under 5-way 5-shot and 5-way 1-shot on *Mini*ImageNet. Table 10 lists the results. It can be seen that our three models achieve lower standard deviations comparing to the three baseline models [23, 42, 44].

**DenseNet backbone performance.** Following the settings of Lichtenstein *et al.* [25], we measure the performance of our approach with DenseNet [19] backbone. Specifically, we insert a FiLM layer after the last BN layer in each dense block of DenseNet-121. The backbone is first pre-trained following the protocol of prior work [49]. The experiments are conducted under 5-way 5-shot and 5-way 1-shot on *Mini*ImageNet and *Tiered*ImageNet. Relative results are listed in Table 4. Our approach outperforms previous state-of-the-art transductive methods by about +1.8%, including RAP-LaplacianShot [17] and TAFSSL [25]. This suggests our approach also contributes to dense connection-based backbones. More details are provided in **Supplementary Material**.

**Comparisons with typical task-adaptive methods.** We compare the few-shot classification performance of our approach with two

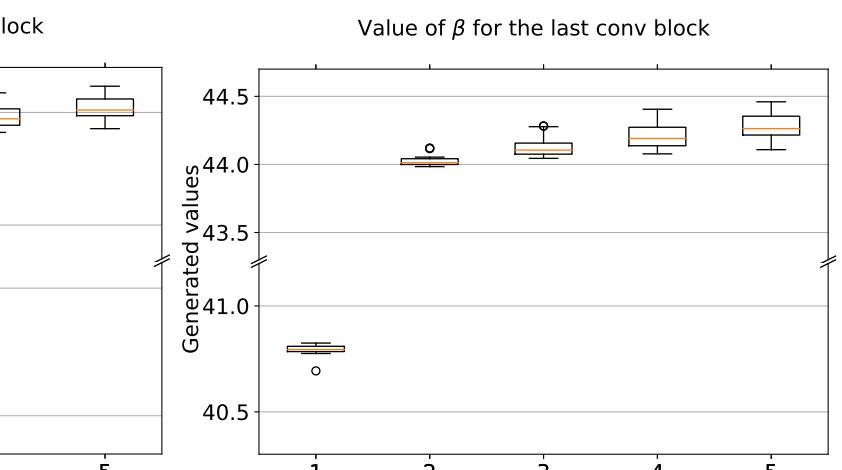

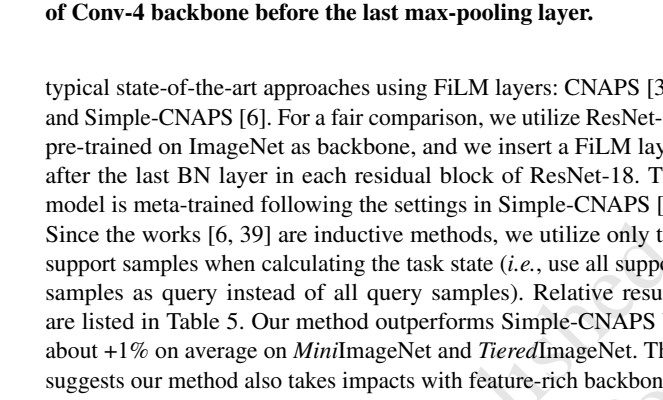

**Figure 4: Visualization of the values of parameters $\gamma$ and $\beta$. We depict the boxplots of $\gamma$ and $\beta$ respectively for the last convolution block of Conv-4 backbone before the last max-pooling layer.**

typical state-of-the-art approaches using FiLM layers: CNAPS [39] and Simple-CNAPS [6]. For a fair comparison, we utilize ResNet-18 pre-trained on ImageNet as backbone, and we insert a FiLM layer after the last BN layer in each residual block of ResNet-18. The model is meta-trained following the settings in Simple-CNAPS [6]. Since the works [6, 39] are inductive methods, we utilize only the support samples when calculating the task state (*i.e.*, use all support samples as query instead of all query samples). Relative results are listed in Table 5. Our method outperforms Simple-CNAPS by about +1% on average on *Mini*ImageNet and *Tiered*ImageNet. This suggests our method also takes impacts with feature-rich backbones, comparing to directly learning parameters of FiLM layers. More details are provided in **Supplementary Material**.

**Cross-domain few-shot classification performance.** Following the splits introduced by Chen *et al.* [8], we meta-train the model on the entire *Mini*ImageNet dataset (*i.e.*, training set, validation set, and testing set), then meta-test the model on the test split of CUB-200-2011 dataset. Relative results are listed in Table 7. We show obvious improvements in our approach under both 1-shot and 5-shot settings compared to other state-of-the-art cross-domain methods. This suggests that the differentiable RL strategy contributes to improving the discriminative ability of task-specific embeddings even when a domain gap exists.

**Generalized few-shot classification performance.** Following the splits of Ye *et al.* [55], we first train the model only on seen classes $\mathbb{S}$ of *Mini*ImageNet dataset under 1-shot 5-way and 5-shot 5-way. Then we evaluate the model with three criteria: UNSEEN evaluates the 5-way classification accuracy only on unseen classes $\mathbb{U}$; SEEN evaluates the 64-way classification accuracy only on seen classes $\mathbb{S}$; COMBINED evaluates the 69-way mixed classification accuracy on both seen and unseen classes. We use the inductive versions of our method in the experiments. Relative results are listed in Table 8. Our method shows about +4.3% and +6.9% performance gains against FEAT [55] and ProtoNet [44] respectively on the 3

evaluation criteria. This suggests that our approach also contributes to not forgetting previously learned class knowledge.

**Visualization.** We depict the statistical distributions, represented as boxplots, of the parameters $\gamma$ and $\beta$ in each time step $t$ across different meta-tasks in Figure 4. We visualize the parameters $\gamma$ and $\beta$ for the last convolution block of Conv-4 backbone. It can be observed that the parameters $\gamma$ and $\beta$ have larger influence on the embedding procedure with longer time steps. Further, it can be seen that the generated parameter values vary among different tasks to a larger extent when the time step goes longer, especially at the last time step $T$. This suggests that the model learns to adapt to each meta-task generally across time steps via differentiable RL. Besides, we plot the curves of validation accuracy on *Mini*ImageNet and *Tiered*ImageNet among our models and ProtoNet. The results are reported in **Supplementary Material**.

## 5 CONCLUSION

This work proposes a generic differentiable RL-based meta optimization strategy for few-shot learning, aiming to explore stable meta-optimization patterns in meta-training. Due to the differentiability of the environment, our approach can avoid the flaws of black-box RL. This approach is a new way of introducing RL strategy into few-shot learning. The approach contains a policy equipped with the memory for learning generalizable optimizations of embedding procedure across unseen tasks. In addition, we propose a task state encoder to adapt information from the query set to the support set for fully exploring inner-task similarities. Our strategy can be easily applied to various backbones, including ConvNet, ResNet, WRN, and DenseNet. Our method also shows promising results on the challenging cross-domain and generalized few-shot classification tasks. Lastly, this work shows the huge benefits of exploring stable patterns in meta-learning, however it has limitations on model complexity. Future work includes studying the model efficiency and applying this approach to multiple downstream tasks.

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
