# OpenReview forum: "Exploring Stable Meta-optimization Patterns via Differentiable Reinforcement Learning for Few-shot Classification"
_acmmm.org/ACMMM/2024/Conference — MM2024 Poster_

### Official Review · Reviewer_Ju72 · 2024-05-03

**Rating:** 5
**Confidence:** 3

**Summary:**

This paper combines reinforcement learning into meta-optimization in few shot learning.  By designing the reward, state and action in reinforcement learning, the proposed approach can generate the parameters \gama and \beta in the BN layer. It is an interesting idea and experiments prove the effectiveness of this approach.

**Strengths:**

1. The work for this kind of combination is interesting,  that adopts generic differentiable RL-based few-shot learning strategy to explore stable meta-optimization patterns for meta-training.
2. The writting is fine for reading.
3. Experiments are ok for me, especially the results on the challenging cross-domain and generalized few-shot classification tasks.

**Limitations:**

1. The gradient updating step T is set as five, which is not a smaller number in many meta-learning approaches.
2. The identification of some variables can be clearer, while some variables do not have explicit meanings.

**Suitability:**

3

---

### Official Review · Reviewer_achM · 2024-05-24

**Rating:** 4
**Confidence:** 3

**Summary:**

This paper presents a differentiable reinforcement learning (RL) strategy for few-shot classification. The aim is to explore stable meta-optimization patterns in meta-training by learning generalizable optimizations for producing task-adaptive embeddings. The proposed strategy models the embedding procedure of feature transformation layers in the meta-learner to optimize the gradient flow implicitly.  The approach is demonstrated to be effective across various backbones and achieves promising results against state-of-the-art transductive methods in few-shot classification.

**Strengths:**

1.The proposed method is solid.

2.The approach is generic and can be easily extended to various backbones, including ConvNet, ResNet, WRN, and DenseNet.

3.The proposed method has been evaluated on three popular datasets and has shown consistent improvement in performance across different backbones.

**Limitations:**

1.To improve the evaluation, the authors should consider including more recent works. The most recent baseline cited is from 2021, which may not adequately reflect the current advancements in the field.

2.The related work section reviews current issues well but lacks details on how this paper addresses them. Please clearly outline the novel contributions and improvements made.

**Suitability:**

2

---

### Official Review · Reviewer_1fL7 · 2024-05-28

**Rating:** 5
**Confidence:** 3

**Summary:**

The paper aims to address the challenges of improving learning stability during meta training. The authors introduce a novel differentiable reinforcement learning strategy RMO to explore stable meta-optimization patterns to enhance the discriminative ability of task-adaptive embeddings. The authors claim that this approach can optimize the gradient flow in the meta-learning process with its differentiable reinforcement learning framework, and thus can make the few-shot learning models to be more robust and efficient.

**Strengths:**

The paper enhances few-shot learning with two key innovations.
First, this paper introduces a new differentiable reinforcement learning framework that can stably manage meta-learners' optimization process. This framework can help maintain stable training across different tasks and address some common issues in traditional RL including the high variance and noisy gradients.
Second, this paper introduces a memory module to learn the continuity between different tasks. This approach is effective when dealing with learning process with limited samples while keeping consistency and stability.
The RMO method improves the model's discriminative performance, and the results of experiments show that RMO consistently outperforms existing state-of-the-art methods in standard benchmarks. Also, the experiments prove that the RMO's ability to dynamically adjust according to the specific needs of certain tasks along with the integrated memory module can enhance the model's performance and reliability, especially in real-world scenarios of limited data and varying tasks.

**Limitations:**

This paper could benefit from several improvements:
First, the article should introduce the principles and characteristics of the FiLM model in the RELATED WORK section before its usage. This would enhance the readability of the paper.
Second, although the authors present the results on multiple datasets, they may consider the balance of the data, which would help to explain the model's robustness on both balanced and imbalanced datasets and further prove the RMO's improvement on stability.
Additionally, integrating various types of attention mechanisms may further enhance the RMO's ability to adapt dynamically to complex tasks, especially in multi-category datasets.
Finally, the Figure 4 needs adjustments for better clarity. There is a wide blank while the key information is folded.

**Suitability:**

3

---

### Official Review · Reviewer_LN5B · 2024-06-10

**Rating:** 6
**Confidence:** 2

**Summary:**

This work proposed a differentiable reinforcement learning framework for few-shot classification tasks. The authors used a Gated Recurrent
Unit (GRU) as the policy module to control the parameters in FiLM layers. The policy module takes task embeddings as input. It also includes a memory module based on task embeddings to retreive the previous task that is most related. The embedding of a task is computed with a "self-attention" module that update embeddings of query samples by attending to support samples. Intensive experiments on different combinations of benchmark datasets and backbone architectures showed significant and consistent improvements over baselines. The authors also conducted complete ablation studies to show the effect brought by each module proposed.

**Strengths:**

+ The paper is written in good quality, with proposed modules explained and illustrated clearly and experiments structured in an organized way.
+ The proposed method, in my opinion is novel. And the empirical results are impressive, with huge improvements over previous methods.
+ The ablation study is complete, providing convincing arguments of the effectiveness of eeach module proposed.

**Limitations:**

- It is good to also add a "baseline" row in Table. 6.
- Some combinations of previous methods were missing in the exp section. For example, the authors of RAP [17] reported RAP+LaplacianShot results on miniImageNet using ResNet-12 backbone, which showed a better 1-shot accuracy of 74.29±0.20%.

**Suitability:**

3

---

### Meta-Review · Area_Chair_6gRr · 2024-06-30

**Recommendation:** Accept (Poster)
**Confidence:** 4

**Metareview:**

This work proposed a differentiable reinforcement learning framework for few-shot classification tasks. After the rebuttal period, this paper received consistent positive ratings. After checking the paper, reviews, and rebuttal document, I decide to accept this paper.

However, as Reviewer Ju72 pointed out, this paper still has some small blemishes in expression. The authors need to fix this in the final version.